# Recent Advances in Nanomedicine: Cutting-Edge Research on Nano-PROTAC Delivery Systems for Cancer Therapy

**DOI:** 10.3390/pharmaceutics17081037

**Published:** 2025-08-10

**Authors:** Xiaoqing Wu, Yueli Shu, Yao Zheng, Peichuan Zhang, Hanwen Cong, Yingpei Zou, Hao Cai, Zhengyu Zha

**Affiliations:** 1Department of Thoracic Surgery and Institute of Thoracic Oncology, Frontiers Science Center for Disease-Related Molecular Network, West China Hospital, Sichuan University, Chengdu 610041, China; wuxiaoqing@wchscu.edu.cn (X.W.); shuyueli2025@wchscu.edu.cn (Y.S.); zhengyao@wchscu.edu.cn (Y.Z.); zhangpeichuan@wchscu.edu.cn (P.Z.); 2West China School of Pharmacy, Sichuan University, Chengdu 610041, China; conghanwen@stu.scu.edu.cn (H.C.); zouyingpei@stu.scu.edu.cn (Y.Z.)

**Keywords:** PROTACs, nanodrug delivery systems, tumor microenvironment-responsive, targeted drug delivery, protein degradation therapy

## Abstract

Proteolysis-targeting chimeras (PROTACs) selectively degrade target proteins by recruiting intracellular E3 ubiquitin ligases, overcoming the limitations of traditional small-molecule inhibitors that merely block protein function. This approach has garnered significant interest in precision cancer therapy. However, the clinical translation of PROTACs is hindered by their typically high molecular weight, poor membrane permeability, and suboptimal pharmacokinetic properties. Nanodrug delivery technologies represent a promising approach to overcome the limitations of PROTACs. By encapsulating, conjugating, or integrating PROTACs into functionalized nanocarriers, these systems can substantially enhance solubility and biostability, enable tumor-targeted and stimuli-responsive delivery, and thereby effectively alleviate the “hook effect” and minimize off-target toxicity. This review systematically outlines the primary design strategies for current nano-PROTAC delivery systems, including physical encapsulation, chemical conjugation, carrier-free self-assembly systems, and intelligent “split-and-mix” delivery platforms. We provide an overview and evaluation of recent advances in diverse nanomaterial carriers—such as lipid-based nanoparticles, polymeric nanoparticles, inorganic nanoparticles, biological carriers, and hybrid nanoparticles—highlighting their synergistic therapeutic potential for PROTACs delivery. The clinical translation prospects of these innovative systems are also discussed. This comprehensive analysis aims to deepen the understanding of this rapidly evolving field, address current challenges and opportunities, promote the advancement of nano-PROTACs, and offer insights into their future development.

## 1. Introduction

In recent years, targeted protein degradation (TPD) technology has rapidly emerged as a research hotspot in drug discovery because of its potential for tackling “undruggable” targets [1]. As a flagship TPD strategy, PROTACs have demonstrated remarkable promise in therapeutic areas such as oncology, leveraging their unique mechanism of action and broad target applicability [2]. PROTACs are heterobifunctional molecules consisting of three distinct components: a protein of interest (POI) ligand, an E3 ubiquitin ligase ligand, and a connecting linker [3]. These molecules operate by simultaneously engaging both the POI and an E3 ubiquitin ligase, inducing the formation of a ternary complex (PROTAC-POI-E3 ligase) [4]. Upon formation of the ternary complex, the POI undergoes polyubiquitination, marking it for proteasomal destruction. The PROTACs, remaining undegraded, is released to catalytically engage additional POI [5]. The hallmark of this process lies in the “event-driven” mechanism of PROTACs, which mediates protein degradation not through sustained occupancy of the protein’s active site, but rather by triggering a cascade of intracellular signaling events [6]. Consequently, PROTACs not only expand the druggable space to include traditionally challenging targets such as transcription factors [7], scaffold proteins, and catalytically inert binding domains [8], but also achieve highly efficient catalytic degradation at substoichiometric doses [9]. PROTACs can overcome cancer drug resistance because they only need to bind to the POI for a short period of time [10]. Even if the POI mutates and reduces its affinity, they can still exert the degrading effect.

Since the initial conceptualization of PROTACs [11], they have attracted significant interest from researchers in the pharmaceutical field. To date, PROTACs have successfully degraded diverse disease-related target proteins, including those implicated in cancer, viral infections, immune disorders, and neurodegenerative diseases [12]. Particularly prominent in anticancer drug development, PROTACs target nuclear receptor proteins, epigenetic regulators, kinases, and other therapeutic targets [13]. Over the past two decades, PROTACs have evolved from peptide-based molecules to small-molecule drugs [14], transitioning from basic drug substances to formulated pharmaceuticals, and advancing from fundamental research to clinical trials. This rapid progression toward commercialization has been accompanied by numerous review articles documenting their development [15,16]. However, as PROTAC research has advanced, certain limitations have emerged, posing challenges for its clinical translation [17]. First, PROTACs typically exhibit large molecular weights (>800 Da) with multiple hydrogen bond donors and acceptors, resulting in high polarity that often violates Lipinski’s rule of five [18]. These properties commonly lead to poor cell membrane permeability and limited oral bioavailability. Second, since PROTACs recruit E3 ligases in a noncovalent manner, some compounds may demonstrate a “hook effect”—a phenomenon where, at high concentrations of PROTAC, a competitive binary complex (E3:PROTAC or PROTAC:POI) is formed instead of a trivalent complex dominate [19]. Furthermore, the tissue-specific expression patterns of E3 ligases contribute to the nonspecific distribution and off-target degradation of PROTACs in vivo, potentially leading to unpredictable toxic effects [20]. To address these limitations, extensive structure-optimization studies are underway, including the development of novel E3 ligase ligands and the screening of optimized linker architectures [21,22]. Structural modifications of PROTACs may address certain limitations, such as improvements in aqueous solubility, stability, and membrane permeability [23]. However, enhancing PROTAC pharmacology solely through medicinal chemistry efforts to achieve effective in vivo efficacy remains a formidable challenge, given the complexity of the tumor microenvironment (TME) [24].

Compared with optimizing PROTACs’ chemical structures to meet in vivo requirements such as tumor targeting and cellular permeability, integrating highly efficient PROTACs with advanced nanodrug delivery systems (nanoDDSs) represents a promising alternative strategy [25]. NanoDDSs offer tunable sizes, functionalizable surfaces, and high drug-loading capacities, providing distinct advantages in enhancing drug stability, bioavailability, and target specificity [26]. Moreover, they have the potential to activate antitumor immune responses [27]. Current nanoplatforms, including liposomes, micelles, inorganic nanoparticles, metal–organic frameworks (MOFs), and biomimetic systems can deliver PROTACs via encapsulation, adsorption, or chemical conjugation [28]. Nano-PROTAC systems utilize diverse construction strategies, such as physical encapsulation within carriers (e.g., liposomes or polymeric nanoparticles) [29], chemically conjugated prodrugs where PROTACs are covalently linked to carrier materials [30], carrier-free self-assembly driven by PROTACs’ hydrophobic interactions with therapeutic agents [31], and “split-and-mix” nanoplatforms that integrate E3 and POI ligands into multivalent polymeric nanostructures [32]. By incorporating stimuli-responsive designs (such as temperature sensitivity, NIR light activation, or enzyme/acid/glutathione responsiveness), these nano-PROTAC delivery systems achieve spatiotemporally controlled release [33]. This ensures that PROTACs activate, release, and degrade target proteins exclusively at disease sites, enhancing therapeutic precision and safety. Notably, nanocarriers not only effectively address PROTACs’ stability limitations and systemic exposure risks, but also leverage their unique nanoscale architectures to enable multimodal combination therapies [34]. These approaches achieve cooperative effects such as synergistic effects between chemotherapy-degradation and phototherapy-immunotherapy. Particularly in cold tumors, therapy-resistant malignancies, and immunosuppressive TMEs, nano-PROTACs have the unique potential to induce immunogenic cell death, enhance antigen presentation, and promote T-cell infiltration [35]. Although clinical translation remains exploratory, promising preclinical studies have demonstrated superior tissue distribution, enhanced therapeutic efficacy, and favorable safety profiles in both solid and hematological tumor models [36]. This preclinical validation paves the way for applications in personalized precision medicine and multimodal combination therapies.

This review systematically elucidates innovative design paradigms and recent advances in nanodrug delivery strategies for enhancing PROTACs’ in vivo delivery efficiency. We examine the construction principles and application scenarios of various delivery systems while evaluating their translational prospects (Figure 1). By comprehensively analyzing nano-PROTAC design strategies, carrier classifications, and therapeutic applications in oncology, we critically examine core challenges in clinical translation and propose countermeasures. This work aims to provide theoretical foundations and practical guidance for developing next-generation targeted protein degradation therapies based on nanoplatforms.

## 2. PROTAC Delivery Strategies via NanoDDSs

NanoDDSs offer unique physicochemical properties and versatile functionality that can address PROTACs’ delivery challenges including poor solubility, suboptimal pharmacokinetics (PK), and inadequate targeting—thereby enhancing bioavailability [37]. First, nanocarriers (e.g., polymeric/lipid nanoparticles, inorganic nanomaterials, and biological carriers) leverage their structural assembly and nanoscale dimensions to improve drug solubility and prolong systemic circulation [38]. Second, targeted surface modification with antibodies, peptides, or small molecules enhances tissue-specific delivery of nano-PROTACs, increasing drug accumulation at disease sites while minimizing off-target protein degradation [39]. Furthermore, nanoDDSs enhance the cellular uptake and transcellular transport of cargo, overcoming membrane permeability limitations [40]. Stimuli-responsive systems (pH/enzyme/light-activated) achieve spatiotemporally controlled drug release at target sites, amplifying therapeutic outcomes [41]. Building upon this established foundation, nano-PROTAC delivery systems have also been the subject of widespread investigation. Currently, the primary delivery strategies for nano-PROTAC systems include physical encapsulation, chemical conjugation, and carrier-free nanodrug delivery systems based on PROTACs self-assembly. Furthermore, owing to the unique structure of PROTACs and their mechanism of mediating interactions between E3 ligases and POIs, a significant strategy for developing nano-PROTAC delivery systems involves the construction of nanodrug delivery systems, with carrier participation, that are capable of inducing proximity between E3 ligases and target proteins [42] (Figure 2 and Table 1).

**Figure 2 pharmaceutics-17-01037-f002:**
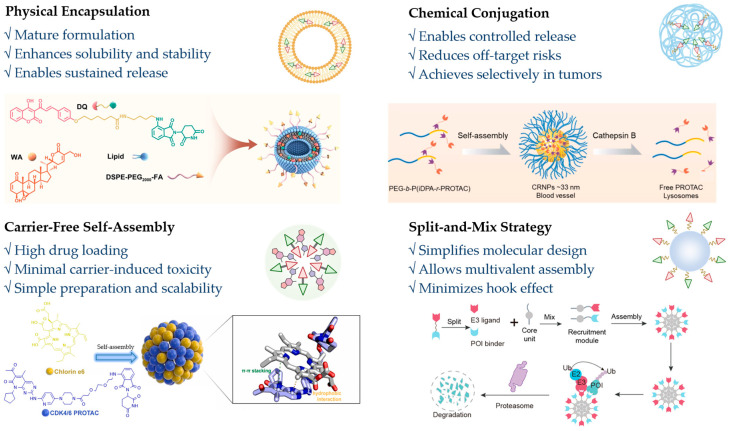
Overview of the advantages of common strategies for constructing nano-PROTAC delivery systems and examples of their applications. The example of physical encapsulation: reprinted with permission from Ref. [43]; Copyright (2023) American Chemical Society. The example of chemical conjugation: reprinted from Ref. [44], Copyright (2023), with permission from Elsevier. The example of carrier-free self-assembly: reprinted from Ref. [45], Copyright (2023), with permission from Elsevier. The example of “split-and-mix”: reprinted with permission from Ref. [46]; Copyright (2023) American Chemical Society.

**Table 1 pharmaceutics-17-01037-t001:** Overview of common strategies for constructing nano-PROTAC delivery systems.

Strategy Type	Key Advantages	Limitations	Suitable Target Types	Representative Examples (Refs Simplified)
Physical encapsulation	Simple process, broad carrier choices; improves solubility and stability; sustained release and passive target	Risk of premature release; limited control over drug localization; relatively lower encapsulation precision	Hydrophobic PROTACs; poorly soluble or unstable PROTACs	PEGylated nanoliposomes encapsulated BRD4 PROTAC [47]; Brain-targeted micelle loaded ARV-825 [48]
Chemical conjugation	Enables stimulus-responsive and tumor-specific release; improves PK; reduces off-target effects	Requires linker design optimization; possible impact on PROTAC activity if linkage affects binding domains	PROTACs targeting TME (e.g., acidic pH, GSH-rich cytosol)	Polymer PROTAC conjugated by pH/cathepsin B sequential responsive release [49]; high level of GSH activated affibody-PROTAC conjugate [50]
Carrier-free self-assembly	High drug loading efficiency; avoids carrier-related toxicity; easier scale-up	Structural dependency on PROTAC properties; physical stability varies; limited to hydrophobic PROTACs	Hydrophobic or amphiphilic PROTACs; suitable for combination with photosensitizers or chemotherapeutics	CDK4/6 PROTAC with Ce6 formed nanoparticles via self-assembly [51]; self-assembly of Ce6 and dBET57 [52]
Split-and-mix	Simplifies synthesis by separating ligands; allows multivalency and spatial control; avoids “hook effect”	Complex system design; requires precise control of assembly in vivo; less established clinically	Protein targets with known E3 and POI ligands that can be independently delivered	Gold nanoparticle-based multi-headed PROTACs (EML4-ALK degraders) [53]; the POI/E3 recruitment modules self-assembled into nanoballs [54]

### 2.1. Physical Encapsulation

Physical encapsulation leverages the inherent properties of nanoDDSs to effectively load and deliver hydrophobic PROTACs. These nanocarriers primarily encapsulate PROTACs within their matrix layers or cavities through hydrophobic interactions, electrostatic adsorption, or core–shell structures [55]. Following targeted delivery to the POI, the PROTAC is subsequently released upon dissociation of the carrier material, thereby exerting its function. Research in this domain has focused primarily on efficient drug loading and controlled release. Commonly employed carriers include liposomes, polymeric micelles, and biotechnological nanocarriers [56]. Moreover, these carriers can be readily functionalized with stimulus-sensitive moieties to form stimuli-responsive nanodrug delivery systems. These systems are specifically activated upon reaching the tumor site, enabling on-demand release [57]. This strategy is frequently employed to initially enhance the bioavailability of PROTACs, which exhibit suboptimal in vivo efficacy or to achieve synergistic effects through co-delivery with other therapeutic modalities, such as phototherapy, radiotherapy (RT), or immunotherapy [51,52]. Zhang et al. engineered folate-modified nanoparticles to co-deliver PROTAC DQ and ROS inducer WA [43]. The nanoparticle surface was further modified with folate to enhance active targeting toward tumor cells. The physical encapsulation strategy holds significant importance in nanomedicine because of its versatility and minimal chemical modification requirements [58]. It is particularly well-suited for delivering chemically sensitive molecules and meeting rapid development needs. However, optimization on the basis of specific drug properties and therapeutic contexts is essential. Physical encapsulation may evolve through synergistic material–biology–clinical integration to simultaneously achieve high-loading, stimuli-triggered release, and reduced toxicity—a triad currently limiting clinical translation. AI-optimized biodegradable carriers and micro-physiological validation systems represent promising enablers for this paradigm shift.

### 2.2. Chemical Conjugation

In contrast to physical encapsulation, the chemical conjugation strategy covalently links PROTACs to functional groups on the surface or within the matrix of nanocarriers, enabling targeted delivery and controlled release [59]. This approach requires that PROTACs possess modifiable functional groups, such as amines, carboxylic acids, or hydroxyl groups, which imposes certain limitations on delivery system design and selection [60]. The design requires balancing stable PROTAC–carrier linkages for delivery integrity with stimuli-responsive release mechanisms. Current stimuli-responsive triggers encompass both exogenous signals (e.g., radiation, magnetic fields, ultrasound, and near-infrared light) and endogenous signals (e.g., pH, cathepsin B, folate receptor expression, and hypoxic conditions) [58]. These modified PROTAC prodrugs can be assembled into nanoparticles with carrier participation, where the active PROTAC is released upon cleavage of the chemical bond under specific conditions. For example, Gao et al. developed POLY-PROTAC micelles via disulfide-linked PROTAC–copolymer conjugates, responding sequentially to MMP-2 and TME [61]. This strategy fundamentally alters the molecular architecture of PROTACs, which can mitigate certain unfavorable factors related to their cellular permeability and PK properties, while simultaneously increasing accumulation at the therapeutic site through stimuli-responsive activation [62]. Furthermore, chemically conjugated nano-PROTAC delivery systems allow the incorporation of targeting molecules, imaging probes, and therapeutic agents (e.g., antibody–PROTAC conjugates, APCs) [63]. Controlled-release nanocarriers for PROTACs offer a promising solution to bridge the dosing gap between cellular assays and animal models. However, this approach presents greater synthetic challenges and necessitates careful consideration of carrier immunogenicity [64]. By addressing linker intelligence, conjugation precision, and translational practicality, chemical conjugation can unlock PROTACs’ full potential while meeting clinical demands. Orthogonal chemistry and optimization of cleavable linkers represents a crucial focus area.

### 2.3. Carrier-Free Self-Assembly

Carrier-free nano-PROTAC delivery systems are engineered through the self-assembly of PROTACs with other functional small molecules. PROTACs possess distinctive structural features, typically including conjugated systems and planar structures based on double bonds [65]. These features enable self-assembly with other molecules (e.g., photosensitizers, chemotherapeutic agents, and inhibitors) via hydrogen bonding, hydrophobic interactions, and π–π stacking, forming stable nanostructures [66]. This design offers high drug loading capacity (approximately 100%), circumvents the toxicity or metabolic burden associated with traditional carriers, and simplifies the manufacturing process [67]. Self-assembled Ce6-PROTAC nanoparticles induce apoptosis and synergistically enhance immunogenic cell death (ICD), reversing tumor immunosuppression [45]. Additionally, light-activated SDNpro nanoparticles (Ce6 + dBET57) enhance photodynamic therapy (PDT) by combining ROS generation with BRD4 degradation [52]. Nevertheless, the clinical translation of carrier-free nanomedicines faces significant challenges, including unpredictable assembly processes, inadequate delivery efficiency, and poor stability [68]. Enhancing the stability and robustness of carrier-free nano-PROTAC delivery systems critically hinges on rational molecular design strategies that strengthen their self-assembly capability and stimuli-responsive traits, coupled with synergistic drug approaches to improve system stability and therapeutic efficacy. By incorporating hydrophobic or stimuli-responsive moieties [69], optimizing linker architectures [70], and leveraging co-assembly for synergistic co-delivery, researchers are progressively expanding the applicability and therapeutic potential of these systems. Molecular dynamics simulations can provide rational guidance for the design of self-assembling PROTACs [71]. Advances in structural biology and AI modeling will enable rational design of self-assembling PROTAC nanostructures with optimized stability and tumor-selective activation. Integrated computational/high-throughput screening (HTS) approaches can streamline development of carrier-free systems featuring high drug loading and microenvironment-responsive release—overcoming key clinical translation barriers [72].

### 2.4. “Split-And-Mix” Nano-PROTACs

Unlike conventional PROTACs, the “split-and-mix” platform offers unique advantages in combinatorial screening efficiency and adaptive target regulation [42]. The core concept of this strategy involves decomposing traditional PROTACs into two functional modules: one bearing the target protein ligand and the other bearing the E3 ligase ligand. Self-assembly, molecular recognition, or external stimuli subsequently drive their association at a specific time and location [46]. This design circumvents the need for preassembly of the complete ternary complex, reduces synthetic complexity, significantly decreases molecular size, and enhances the loading flexibility and systemic stability of the nanoplatform [73]. For example, the use of nanogold carriers to separately load POI ligands and E3 ligase ligands can overcome the druggability hurdles of PROTACs, improving their solubility, stability, and half-life [53]. Similarly, packaging bioorthogonal components within integrin αvβ3-engineered nanoparticles or employing tumor cell-specific sulfatase-responsive peptides for in situ assembly ensures spatiotemporally controlled co-localization of the POI ligand and E3 ligase ligand, leading to the in situ generation of active PROTACs [74]. Delivering these ligands separately avoids the low permeability and rapid clearance issues associated with large bifunctional molecules. Different delivery modules can be independently designed to target distinct tissues or cell types, optimizing biodistribution [75]. This approach not only preserves the PROTAC degradation mechanism, but also confers high conformational freedom and spatial control, mitigating activity loss due to the “hook effect” [32]. Additionally, the “split-and-mix” strategy opens avenues for multivalent and multipoint cooperative degradation. For example, studies have arranged combinations of the POI recruitment modules/E3 recruitment modules at different ratios and self-assembly into nanoballs to increase the binding stability and degradation efficiency of the ternary complex [54]. Overall, “split-and-mix” nano-PROTACs, through their modular functionalization, versatile loading, and controllable triggering mechanisms, effectively address the drawbacks of conventional PROTACs, such as synthetic complexity, conformational constraints, and suboptimal degradation efficiency caused by the “hook effect”. However, PROTAC efficacy depends critically on maintaining optimal stoichiometry and local concentration of its modules. Inadequate tumor co-localization or subthreshold concentrations impair degradation efficiency, while uncontrolled assembly kinetics may cause delayed action or premature release—key challenges limiting therapeutic outcomes. With further advancements in molecular self-assembly technologies, biomimetic materials, and target recognition strategies, such systems hold significant promise as pivotal components in advancing the clinical translation of nano-PROTACs [76]. Key focus areas should include developing stimuli-responsive assembly/disassembly mechanisms for spatiotemporal control, engineering biohybrid carriers that leverage natural trafficking pathways, and implementing machine learning to optimize PROTAC ternary complex kinetics within TME.

## 3. Recent Research Advances in Nano-PROTAC Delivery Systems

With the successful application of nanoplatforms for small-molecule drugs, nucleic acid therapeutics, and protein-based pharmaceuticals, researchers have progressively integrated nanocarrier technology into PROTACs delivery [77]. Implementation of these nanoplatform strategies using smart materials facilitates clinical translation of nano-PROTACs by addressing key challenges in stability, biodistribution, and degradation efficiency [78]. Furthermore, emerging evidence indicates that certain nanomaterials possess intrinsic synergistic properties with PROTACs. For example, specific metal ions or oxide materials can modulate the tumor microenvironment, thereby augmenting the activity of the ubiquitin-proteasome pathway induced by PROTACs [49]. Certain natural polymers or biomimetic membrane structures enhance systemic immune evasion capabilities and blood stability, consequently prolonging the in vivo half-life of PROTACs [79]. Consequently, nano-PROTAC systems have emerged as novel platforms for combinatorial antitumor therapy. They are employed not only for single-protein degradation, but also in combination with chemotherapy, immunotherapy, photothermal therapy, and other modalities [52]. This synergistic approach enhances the therapeutic response and delays the onset of drug resistance.

Based on our analysis of the literature data, the number of nano-PROTACs has shown a substantial increase over the past five years. Leveraging established nano-PROTACs delivery strategies, diverse carriers exhibit distinct advantages, evolving from rudimentary encapsulation to advanced multifunctional and stimuli-responsive platforms. To provide a clear visualization of current advancements and technological evolution, we systematically compiled recently reported nano-PROTAC studies into a comprehensive table (Table 2). The subsequent sections provide a systematic classification and critical analysis of current research, organized by both nanomaterial architecture and functional performance metrics.

### 3.1. Application of Lipid-Based Nanoparticles in PROTACs Delivery

Lipid-based nanoparticles (LBNPs) are hierarchical nanoDDSs primarily composed of neutral phospholipids, cholesterol, surfactants, and functionalized lipids in precisely optimized ratios [94]. Their canonical structure consists of spherical vesicles (40–200 nm diameter) featuring at least one aqueous core enveloped by one or more concentric lipid bilayers. The distinctive “lipid–aqueous interface” architecture of LBNPs confers exceptional capabilities for penetrating biological barriers and accommodating diverse therapeutic payloads. LBNPs, particularly liposomes and lipid nanoparticles (LNPs), have been extensively explored for drug delivery applications. Liposomes, featuring phospholipid bilayers around an aqueous core, effectively encapsulate both hydrophilic and hydrophobic drugs [95]. They enable passive diffusion or active targeted release through surface modifications, widely used in chemotherapy [96]. In contrast, LNPs represent a newer generation of lipid-based carriers, generally composed of ionizable lipids, structural lipids, and PEGylated components [97]. These systems form compact lipid cores capable of efficiently encapsulating and delivering nucleic acids. The 1995 approval of Doxil^®^ marked a pivotal milestone for liposome-based cancer nanomedicine [95], with subsequent approvals of DepoDur^®^ (2004) and Exparel^®^ (2011) further validating their clinical translatability [98]. Notably, LNPs revolutionized nucleic acid delivery via COVID-19 mRNA vaccines [99], while lipid modifications enable tissue-specific targeting to reduce toxicity [100].

PEGylated liposomes significantly enhance PROTAC delivery by prolonging systemic circulation through RES evasion while enabling synergistic co-delivery strategies [47,101,102]. These include combining PROTACs (e.g., ARV825) with chemotherapeutics like docetaxel (DTX) to improve solubility and sustain release [103], or with siRNAs to boost cellular uptake and degradation efficiency [104]. Lipid nanoformulations effectively address critical PROTAC limitations [105], enhancing solubility, extending half-life, and improving bioavailability, demonstrating strong clinical translation potential [29]. NanoDDSs enhance tumor targeting while minimizing systemic toxicity, as demonstrated by Wang’s team using ROS-responsive lipid nanoparticles (dGPX4@401-TK-12, ~200 nm) for GPX4 degradation-induced ferroptosis (Figure 3A) [80,106]. These nanoparticles exhibited H_2_O_2_-triggered release (>60% over 12 h in vitro), leveraging tumor-specific ROS levels for precise dGPX4 delivery. The system showed enhanced therapeutic efficacy and reduced toxicity in vivo compared to free PROTACs, highlighting the dual advantages of tumor-selective activation and payload stabilization. Furthermore, Patel et al. developed galactose-modified nanoliposome (GALARV, 93.8 ± 10.1 nm) for liver-targeted BRD4 degrader (ARV) delivery via ASGPR recognition [81,107]. GALARV enhanced ARV accumulation in HCC cells compared to non-targeted liposomes (~3-fold greater) and free drug (~4.5-fold greater), demonstrating superior cytotoxicity and pro-apoptotic effects. Wang et al. developed Lip@inS3-TEG-VL-TK liposomes to overcome cancer stem cell (CSC)-mediated HCC resistance by delivering a STAT3-targeting PROTAC prodrug [82]. This system enhances PROTAC stability against enzymatic degradation, and enables TME-responsive activation (e.g., ROS-triggered release). Leveraging EPR-mediated passive targeting and potential galactose modification, it achieves tumor-specific accumulation while overcoming typical CSC resistance mechanisms. To overcome resistance to targeted inhibitors in lung cancer, Patel et al. developed GE11-functionalized LNPs (T-BEPRO) co-encapsulating EGFR/BRD4-targeting PROTACs [108]. This system enhanced tumor-specific uptake and growth inhibition while minimizing off-target effects, demonstrating how surface modifications enable precise tumor recognition and drug delivery.

Recent advances in multifunctional LBNPs have enabled innovative PROTAC delivery strategies for synergistic antitumor therapy. Yao et al. designed a thermo/NIR-responsive nanoplatform with a DPPC/DSPE-PEG_2000_-NH_2_ shell encapsulating BP and surface-conjugated PROTAC-peptide prodrug (DPCP) [109]. Tumor-localized CatB activates the PROTAC for sustained DHODH degradation, while NIR-triggered BP release inhibits GPX4 via selenocysteine binding, synergistically inducing ferroptosis through dual-pathway targeting. This process combines with the tumor-specific activation of the pro-PROTAC via highly expressed enzymes to increase its specificity of action. Huang et al. developed L@NBMZ liposomes co-encapsulating BRD4-targeting PROTAC MZ1 and photosensitizer NBSEt for light-activated combination therapy [83]. The system demonstrated potent tumor accumulation and excellent biosafety, achieving complete BRD4 degradation with concomitant caspase-3 activation (Figure 3B). Remarkably, this approach led to complete tumor eradication within 21 days without recurrence, showcasing the clinical potential of photoactivatable nano-PROTAC therapeutics.

LBNPs, especially liposomes, have become a cornerstone for PROTAC delivery research, successfully expanding from conventional small-molecule PROTACs to more complex bioengineered variants. Recent breakthroughs demonstrate how liposomes can effectively deliver engineered bioPROTAC templates through strategic complexation with charged or ionizable lipids, facilitating efficient cell membrane penetration and subsequent cytosolic release of these biological degraders [110]. Beyond single-agent delivery, liposomes show remarkable versatility as combination therapy platforms. A prime example includes co-loading hydrophobic PROTACs with siRNA and chemotherapeutic drugs, creating synergistic systems that simultaneously leverage protein degradation and gene silencing to combat challenging therapeutic obstacles like chemotherapy resistance and tumor immune evasion [111]. Future development should prioritize: (i) novel lipid chemistries to enhance endosomal escape, (ii) tumor-specific targeting moieties to improve therapeutic indices (e.g., folate-modified systems showing 3-5-fold tumor accumulation), and (iii) rational combination with immunotherapies. The clinical translation of these systems will ultimately depend on resolving the fundamental trade-off between drug loading capacity and nanoparticle stability during manufacturing.

### 3.2. Application of Polymeric Nanoparticles in PROTACs Delivery

Polymeric nanoparticles (PNPs) are composed of natural or synthetic polymers and self-assemble in aqueous solutions to form a core–shell structure [112]. Drugs can be encapsulated within the nanoparticle core or conjugated to the polymer backbone [113]. Drug release occurs upon specific stimuli that trigger either the disintegration of the polymer matrix or the cleavage of linkers. To date, nearly 20 nanomedicines based on PNPs have received FDA approval. Examples include Copaxone^®^ [84], Neulasta^®^ [114], PLEGRIDY^®^ [115], and ADYNOVATE [116].

Building on their success in delivering chemotherapeutics and nucleic acids, PNPs now show promise for overcoming PROTAC delivery challenges. Patel et al. developed PLGA-PEG PNPs encapsulating BRD4-targeting ARV-825, achieving stable 90 nm nanoparticles with extended half-life and enhanced efficacy [117]. The modular PNP platform enables further tumor-targeting optimization through antibody conjugation [118] or ligand modification (e.g., cRGD [119]), demonstrating versatile potential for precision PROTAC delivery. Huang et al. developed integrin αVβ3-targeted PLGA nanoparticles delivering bioorthogonal PROTAC prodrug MZ1-O, which achieves tumor-specific activation via tetrazine (Tz)-loaded microneedles [120] (Figure 4A). This system demonstrated concentration-dependent BRD4 degradation and induced significant apoptosis upon Tz-triggered MZ1-O activation, establishing a precision strategy for spatiotemporal control of protein degradation. The approach leverages bioorthogonal chemistry to overcome off-target effects while maintaining therapeutic efficacy. Substance P (SP)-functionalized nanoformulations exploit NK-1R overexpression in gliomas to achieve targeted PROTAC delivery, demonstrating superior antitumor efficacy over non-targeted systems [48]. These SP-modified micelles enhance tumor-specific drug accumulation while minimizing systemic exposure, as evidenced by significant tumor regression (reduced volume/weight) without observable toxicity (stable body weight), establishing a precision platform for PROTAC-based glioma therapy. The acidic TME drives selective disassembly of pH-sensitive MPEG-PAE micelles co-loaded with ERD308 and palbociclib, achieving 87% tumor-specific payload release and 48.6-fold enhanced tumor accumulation versus non-responsive systems [121]. This strategy simultaneously reduces hepatic exposure by 40%, prolonging therapeutic duration while minimizing off-target toxicity—demonstrating that TME-responsive PNPs can optimize PROTAC delivery precision. Furthermore, GSH-responsive PDSA nanoparticles encapsulating ARV-771 demonstrate dual advantages: DSPE-PEG-5000 modification enhances PK and tumor accumulation, while the disulfide backbone enables redox-triggered release [122]. This system outperforms free ARV-771 with significantly improved BRD4 degradation efficiency and antitumor efficacy, validating nanomaterials innovation as a strategic approach to amplify PROTAC therapeutic potential. In addition to GSH, CatB-responsive nanoDDSs demonstrate remarkable tumor-targeting precision, as exemplified by NIR-activated semiconducting PNPs that leverage CatB overexpression for selective PROTAC release [123,124]. These systems synergize effectively with PDT/sonodynamic therapy (SDT) modalities [35,125], while multi-stimuli-responsive designs further enhance spatiotemporal control, showcasing the versatility of biomarker-driven PROTAC delivery platforms. Combining multiple stimuli can provide superior spatiotemporal control over PROTACs activation and delivery. Yang et al.’s dual-responsive PSRNs overcome key PROTAC delivery barriers through sequential size transformation [126]: stable 40 nm circulation enables tumor accumulation via EPR, while acidic TME-triggered disassembly (<10 nm) enhances penetration. Subsequent CatB-mediated cleavage of GFLG linkers in lysosomes achieves precise intracellular PROTAC release, yielding 2.6–3.0-fold greater CDK4/6 degradation versus free PROTACs—demonstrating that multi-stimuli systems can optimize spatiotemporal control of protein degradation.

Recent advances demonstrate that covalent PROTAC–polymer conjugates can self-assemble into stimulus-responsive micelles, enabling tumor-targeted delivery and controlled release of active PROTACs via TME or external triggers (e.g., NIR-cleavable linkers) [127,128]. These systems synergize protein degradation with phototherapy [30,65] while concurrently overcoming immune suppression through targeted degradation of immunoregulatory proteins—significantly amplifying antitumor efficacy [129,130,131]. Similarly, Jiang et al.’s bilayer MN patch integrates acid/light-dual responsive PPT7 NPs with oxygen-generating BM nanoparticles for enhanced GBM therapy [49]. The outer layer’s PPT7 NPs release PROTAC prodrugs and photosensitizers upon acidic activation, while NIR-triggered PDT generates cytotoxic ^1^O_2_ for prodrug activation. Simultaneously, the inner BM layer alleviates tumor hypoxia through H_2_O_2_ catalysis, synergistically enhancing both PDT efficacy and PROTAC functionality—demonstrating a smart combinatorial approach to overcome therapeutic resistance in GBM. Furthermore, these X-ray-activated micelles exemplify that stimulus-responsive PROTAC systems can potentiate conventional therapies (e.g., radiotherapy) [85,132,133], while modular designs enable tumor-targeted or organelle-specific delivery—expanding the therapeutic window for protein degradation strategies [134,135]. The pH/enzyme cascade-responsive CRNPs utilize an ultra-pH-sensitive nanoplatform with CatB-cleavable linkers to achieve tumor-selective PROTAC release, transitioning from stable 33 nm nanoparticles under physiological conditions to activated degraders in the acidic TME (Figure 4B) [44]. This dual-responsive system demonstrates prolonged circulation and remarkable efficacy (94% tumor inhibition in Karpas299 models) while maintaining excellent biocompatibility, showcasing the transformative potential of stimuli-responsive PROTAC delivery for precision oncology.

Overall, PNPs represent a transformative platform for PROTAC delivery, addressing critical pharmacological challenges through their high loading capacity and tunable release kinetics—particularly advantageous for chronic conditions like cancer and inflammatory diseases [136]. By overcoming PROTACs’ inherent limitations in solubility and stability, PNPs enhance bioavailability while enabling precise targeting across biological barriers. The integration of stimuli-responsive mechanisms further refines therapeutic precision, maximizing on-site activation while minimizing systemic toxicity [123]. Importantly, their modular design facilitates synergistic combinations with chemo-/immunotherapies, expanding treatment paradigms [124]. However, clinical translation demands rigorous safety evaluation, as even biodegradable polymers must demonstrate complete metabolic clearance of degradation byproducts to mitigate long-term toxicity risks. This balance of efficacy and safety positions PNPs as a promising yet cautiously optimizable strategy for advancing PROTAC therapeutics.

### 3.3. Application of Inorganic Nanoparticles in PROTACs Delivery

Inorganic nanoparticles (INPs) have demonstrated successes in imaging and treatment of tumors [137]. INPs are nanostructures composed of inorganic materials such as metals (e.g., gold and silver), metal oxides (e.g., silica and iron oxide), quantum dots (e.g., CdSe), or carbon-based materials (e.g., graphene) [138]. In contrast to organic nanoparticles, INPs emerged in the late 20th century and their biomedical applications are still being actively explored. INPs exhibit unique advantages for biomedical applications owing to their distinctive physicochemical properties (e.g., magnetic/thermal/catalytic activities) and versatile surface engineering capability [139]. Their rigid nanostructure minimizes premature drug leakage while enabling precise imaging-guided therapeutic delivery, making them particularly valuable for cancer theranostics and controlled drug release applications.

Ultrasmall gold–peptide nanocomposites (<10 nm) were developed to overcome PROTAC peptide limitations, where PEI-mediated one-step synthesis with chloroauric acid enabled efficient AR-pep-PROTAC loading and cationic surface engineering [140]. This system enhanced tumor accumulation, prolonged half-life, and improved cellular penetration, demonstrating potent protein degradation and antitumor efficacy without significant toxicity in preclinical models. Similarly, gold-nanoparticle-delivered GPX4-targeting PROTAC (Au-PGPD) effectively induced selective GPX4 degradation in acute lymphoblastic leukemia (ALL) cells while sparing normal cells, demonstrating potent antitumor activity with minimal off-target effects—highlighting the potential of noble metal platforms for precision protein degradation therapeutics [86]. Li et al.’s AuPLs integrates gold nanocubes with phase-change materials (LA/SA) to co-deliver a PROTAC prodrug (dBET1) and Pd catalyst, achieving spatiotemporally controlled activation (Figure 5A) [87]. NIR-triggered PCM melting synchronously releases both components, enabling tumor-localized bioorthogonal deprotection that restores dBET1 activity while minimizing off-target effects—demonstrating 3-fold enhanced tumor accumulation versus conventional administration in murine models. In addition to the gold-based systems discussed above, mesoporous silica nanoparticles (MSNs) have also been developed for PROTACs delivery [141,142]. Specifically, Yu et al. designed a photocaged PROTAC prodrug (phoBET1) and encapsulated it within upconversion nanoparticle (UCNP)-based mesoporous silica nanoparticles (UMSNs) [88]. The UMSNs@phoBET1 nanocage represents a breakthrough in precision PROTAC delivery by utilizing upconversion nanoparticles (UCNPs) to convert 980 nm NIR light into localized UV activation, enabling deep-tissue BRD4 degradation with >80% tumor inhibition. This innovative approach overcomes the critical limitations of conventional UV-activated systems by providing superior tissue penetration while minimizing off-target effects, establishing a new paradigm for spatially controlled protein degradation therapeutics.

INPs, including gold nanoparticles and mesoporous silica, have emerged as promising platforms for PROTAC delivery, offering distinct advantages in drug loading capacity and surface functionalization. Notably, the incorporation of metal ions (e.g., Mn^2+^) can further potentiate therapeutic efficacy by enhancing antigen presentation, enabling synergistic combinations with immunotherapy [143]. However, the clinical translation of INP-based PROTAC delivery systems faces considerable challenges, particularly regarding their non-biodegradability and potential long-term toxicity [144]. To address these limitations, future efforts should focus on three key directions: (i) developing biodegradable inorganic materials (e.g., calcium phosphate nanoparticles), (ii) optimizing stimulus-responsive release mechanisms (e.g., ultrasound or ROS activation), and (iii) improving biocompatibility through hybrid designs such as exosome-coated or organic–inorganic composite carriers. Beyond delivery applications, the unique physicochemical properties of INPs—such as their intrinsic imaging contrast and catalytic activity—position them as ideal candidates for theranostic PROTAC platforms, capable of integrating real-time monitoring with targeted protein degradation.

### 3.4. Application of Biological Carriers in PROTACs Delivery

Biological carriers, leveraging their inherent biocompatibility, low immunogenicity, and functional programmability, are widely utilized in the development of delivery systems, with the current focus primarily on albumin, antibodies, nucleic acids, and exosomes [145]. In recent years, deep integration with nanodrug delivery technologies has enabled a transformative upgrade from their natural properties to engineered functionalities [146]. Nanotechnology facilitates precise control over carrier size, surface charge, and topological structure, significantly increasing the delivery efficiency and targeting precision of biological carriers. Moreover, nanoengineering endows these materials with smart stimuli-responsive properties and multifunctional integration capabilities [147]. This synergy not only enhances and extends the intrinsic advantages of biological carriers (such as low immunogenicity and transmembrane transport efficiency), but also overcomes the limitations of traditional delivery systems (e.g., low drug loading capacity and off-target risks) through nanoscale functionalization [147].

Albumin-based nanoparticles demonstrate exceptional potential for PROTAC delivery, exemplified by BP3@HSA NPs enhancing HSP90-PROTAC pharmacokinetics and efficacy [148,149]. A breakthrough strategy conjugates ARV-771 to ECMal linkers that spontaneously bind serum albumin, achieving tumor-selective PROTAC release via esterase cleavage—resulting in 16.3-fold greater tumor accumulation and 5.3-fold enhanced therapeutic index versus free PROTAC while eliminating systemic toxicity (Figure 6A) [89]. This albumin-hijacking approach leverages tumor-specific esterase overexpression to create a powerful dichotomy: normal tissue protection coupled with potent intratumoral protein degradation. Ferritin PROTACs spontaneously form tumor-targeting albumin nanoparticles via oleic acid binding, demonstrating potent MCL-1 degradation and enabling NIR-II image-guided surgery [150,151]. Additionally, several biomacromolecules (such as antibodies and peptides [152,153]) have been employed to modify conventional PROTACs to enhance their targeting ability. Antibodies are frequently utilized as targeting ligands because of their high affinity and specificity for targets, which can reduce off-target effects [154,155,156]. Similarly, HER2-targeted PROTAC nanoparticles (ZHER2:342-MZ1) leverage affibody-mediated endocytosis and GSH-responsive release, achieving tumor-specific degradation while minimizing off-target effects [50]. These biomolecule–PROTAC hybrids exemplify how endogenous transport mechanisms can be harnessed for precision oncology applications. Recently, the AS1411 aptamer has emerged as a promising tumor-targeting ligand due to its high affinity for nucleolina—protein overexpressed on cancer cells but minimally present in normal tissues [157,158,159]. This unique binding specificity, combined with the aptamer’s low immunogenicity and superior tissue penetration, enables highly selective PROTAC delivery to malignant cells while sparing healthy tissue. He et al. conjugated a BET-targeting PROTAC to AS1411 aptamer via a cleavable linker [160]. Compared with unmodified BET PROTACs, this conjugate demonstrated superior tumor-targeting capability in MCF-7 xenograft models, enhancing BET degradation and antitumor efficacy in vivo while reducing toxicity. As programmable and modifiable nanomaterials, DNA tetrahedrons enable precise molecular assembly, selective recognition, and flexible regulation of intermolecular distances, making them ideal linkers. Li et al. employed a DNA tetrahedron as a structural scaffold to simultaneously integrate three functional modules: (i) tumor cell-targeting ligands, (ii) E3 ligase binders, and (iii) multiple POI-targeting warheads [161] (Figure 6B). The resulting multivalent PROTAC (AS-TD2-PRO) mediated the potent and sustained degradation of the STAT3 protein. Compared with conventional bivalent PROTACs, AS-TD2-PRO features dual STAT3-recognition modules that demonstrate enhanced tumor-specific targeting and degradation efficiency. This work addresses two critical limitations of traditional PROTACs: suboptimal efficacy and off-target toxicity.

By combining DNA nanotechnology with PROTAC design, a highly precise, programmable strategy for protein degradation therapeutics can be established. Camel milk-derived exosomes (CMEs) serve as a versatile delivery platform for anticancer agents such as ARV-825, improving drug solubility, permeability, and bioavailability [90]. Furthermore, the inherent biocompatibility and natural origin of CMEs reduce the immunogenicity risks and adverse reactions associated with conventional drug delivery systems, increasing their potential for clinical translation. Biological carriers are revolutionizing drug delivery through their inherent molecular recognition capabilities and programmable properties, enabling the development of precision medicines with enhanced therapeutic intelligence.

### 3.5. Application of Hybrid Nanoparticles in PROTACs Delivery

Hybrid nanomaterials integrate multiple functional advantages for oncological applications, including stimulus-responsive behavior, high target specificity, and enhanced biological barrier penetration capabilities [162]. In recent studies on nanoDDSs, hybrid nanocarriers (such as organic–inorganic hybrids, cell membrane-coated NPs, and MOFs) have emerged as pivotal strategies for constructing high-performance PROTAC delivery platforms [163]. Organic–inorganic hybrid carriers synergistically combine the advantages of both components to revolutionize PROTAC delivery. He et al.’s Au-PEI system demonstrates enhanced tumor accumulation through pH-responsive PEI protonation and GSH-triggered release [91], while Zhu et al.’s MOF platform co-loads PROTACs and photosensitizers for combinatorial therapy [92]. These hybrid platforms synergistically combine exceptional drug loading, microenvironment-responsive release kinetics, and molecular targeting precision with integrated diagnostic capabilities, marking a transformative advancement in targeted protein degradation strategies.

Cell membrane-coated (CMC) hybrid systems synergize biological advantages (low immunogenicity, targeting capability) with synthetic benefits (high payload, controlled release), enabling breakthrough PROTAC delivery with enhanced targeting, prolonged circulation, and multifunctional integration for improved clinical translation [79]. CMC nanoparticles leverage specialized biological functions for enhanced PROTAC delivery—erythrocyte membranes evade immune clearance [164], while tumor cell membranes enable homotypic tumor targeting [165], collectively improving permeability [166], degradation efficiency, and enabling synergistic chemo-/immunotherapy combinations [167,168]. Xu et al.’s M@TP nanovesicles combine GSH-responsive liposomes with glioma cell membranes, achieving dual homotypic tumor targeting and enhanced BBB penetration for potent GBM therapy, as confirmed by TEM characterization and superior tumor accumulation versus conventional LNPs (Figure 7A) [93]. CMC polymeric systems synergize synthetic polymer advantages (high stability/drug-loading) with biological membrane benefits (biocompatibility/targeting), as demonstrated by Guan et al.’s dual-targeted nanoparticles that combined pH/GSH-responsive dBET6 release with homologous membrane camouflage and cRGD modification for enhanced tumor accumulation [169]. Recent studies demonstrate that tumor cell membrane-coated Si-Fe nanoparticles enable homologous cancer targeting and PROTAC delivery (FAK-P/BRD-P), while exhibiting optimized colloidal stability, prolonged circulation, and reduced macrophage clearance—validating their clinical potential for precision oncology applications (Figure 7B) [163].

Overall, owing to their structural diversity, tunable functionality, and highly integrated capabilities, hybrid nanoDDSs offer highly customizable solutions for the efficient delivery of PROTAC-based drugs. Looking ahead, the integration of AI-guided material design, optimization of biodegradable modules, and modulation of TME-responsive mechanisms are expected to further broaden their adaptability across diverse tumor types. This advancement will offer robust support for the clinical translation of nano-PROTACs.

## 4. Clinical Translation Challenges and Prospects for Nano-PROTAC Delivery Systems

Nano-PROTAC delivery systems merge nanomaterial versatility with PROTAC precision for targeted protein degradation, showing strong preclinical potential. However, clinical translation faces key challenges in carrier optimization, safety assessment, therapeutic efficacy, and scalable production.

First, the complex structure of nano-PROTAC delivery systems (liposomes, polymers, MOFs) creates manufacturing challenges, requiring multi-step processes, high-purity materials, and strict quality control that hinder scalable production [170]. Nano-PROTAC formulations face stability challenges in physiological conditions, where protein corona formation, pH shifts, and enzymatic degradation disrupt structural integrity and controlled drug release [171]. Second, biocompatibility and safety concerns of nano-PROTACs, including potential immune activation and long-term bioaccumulation risks, require further preclinical validation despite mitigation strategies like PEGylation and biodegradable materials. Current understanding of their in vivo pharmacokinetics remains limited. Current nano-PROTAC research remains largely limited to in vitro and animal studies, lacking comprehensive human-relevant data on metabolism, targeting efficiency, and systemic toxicity [172]. In addition, nanoformulation often results in increased particle size—commonly in the range of 50–200 nm—which may hinder effective tumor penetration, especially in dense or poorly vascularized tumors. Recent studies have emphasized that nanoparticles smaller than 50 nm show superior interstitial diffusion and deeper tumor infiltration [173]. To overcome this, researchers are exploring size-optimized or stimuli-responsive shrinking nano-PROTAC systems to enhance intratumoral distribution and therapeutic efficacy. Smart nano-PROTACs require (i) TME-responsive release to maintain degradation efficacy, (ii) dose control to prevent the hook effect, and (iii) standardized characterization protocols—currently unmet needs for clinical advancement. Current nano-PROTAC development lacks standardized evaluation platforms, with inconsistent assessment of loading capacity, release kinetics, targeting efficiency, and safety, while PROTAC diversity further complicates carrier compatibility and regulatory approval.

Despite these challenges, research on nano-PROTACs continues to demonstrate immense promise. Recent advances in biomaterials science, TME modulation strategies, and HTS platforms have spurred the development of increasingly sophisticated stimuli-responsive, targeted, and multifunctional integrated nanoplatforms. These innovations significantly expand the therapeutic potential of PROTACs in cancer treatment. Simultaneously, several technological approaches have advanced to preclinical development, laying crucial groundwork for future translation. Furthermore, nano-PROTACs show promise when combined with immunotherapy, phototherapy, and chemotherapy for personalized cancer treatment. Therefore, despite the significant translational hurdles, nano-PROTAC systems hold considerable potential in oncology. By optimizing nanocarrier design, enhancing safety evaluation, and improving drug delivery efficiency, we can advance the clinical application of nano-PROTAC systems. Furthermore, refining regulatory standards, rationally designing clinical trials, reducing production costs, and fostering multidisciplinary collaboration will help provide cancer patients with more precise and efficient treatment options.

## 5. Summary and Outlook

PROTACs revolutionize cancer therapy by degrading “undruggable” targets via E3 ligase recruitment, yet face clinical challenges due to poor bioavailability and stability. NanoDDSs overcome these limitations by enhancing PROTAC targeting, stability, and enabling novel therapeutic modalities. Recent advances in nano-PROTACs have yielded diverse design strategies (encapsulation/conjugation/carrier-free/“split and mix”) and material platforms that collectively enhance PROTAC solubility, stability, and tumor accumulation. These systems enable precise microenvironment-responsive release, targeted delivery, and synergistic combination therapies, significantly expanding their therapeutic potential across cancer. LBNPs exhibit exceptional tumor-targeting capabilities through EPR effects and ligand-mediated active targeting, achieving 3-5-fold greater accumulation compared to free drugs. Polymeric carriers provide an ideal platform for tumor-specific prodrug activation due to their precisely tunable chemical structures that respond to TME stimuli (e.g., pH, enzymes). INPs offer unique theranostic advantages by combining contrast-enhanced imaging (e.g., MRI/CT) with stimuli-triggered drug release capabilities. CMC systems significantly prolong circulation half-life through inherited biological functions while maintaining targeting specificity. Collectively, these nanoplatforms address distinct aspects of PROTACs delivery from spatial control (PNPs), real-time monitoring (INPs), to systemic persistence (CMC)—demonstrating complementary strengths for clinical translation.

In summary, nano-PROTAC delivery systems offer novel therapeutic options for cancer by addressing the inherent delivery challenges of conventional PROTACs. Future development may prioritize rigorous biosafety assessment, scalable manufacturing processes, and standardization of clinical translation pathways. Concurrently, deepening the understanding of their pharmacological mechanisms and establishing dynamic monitoring techniques are crucial for advancing these systems from bench to bedside. Driven by continuous design optimization and innovative strategies, nano-PROTACs hold transformative potential in oncology, accelerating the clinical realization of next-generation precision protein degradation therapies.

## Figures and Tables

**Figure 1 pharmaceutics-17-01037-f001:**
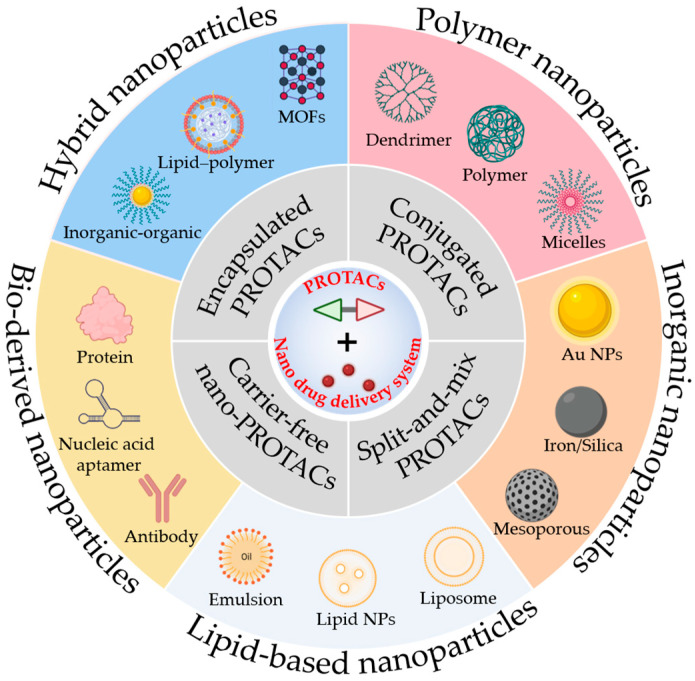
Common strategies and delivery carrier classifications for constructing nano-PROTAC delivery systems. The construction strategies for nanodrug delivery systems of PROTACs in the inner circle can be rationally designed and implemented based on the diverse nanocarrier platforms outlined in the outer circle, encompassing polymer-based, inorganic, lipid-based, biologically derived, and hybrid nanoparticles. This figure includes a portion generated from BioRender.

**Figure 3 pharmaceutics-17-01037-f003:**
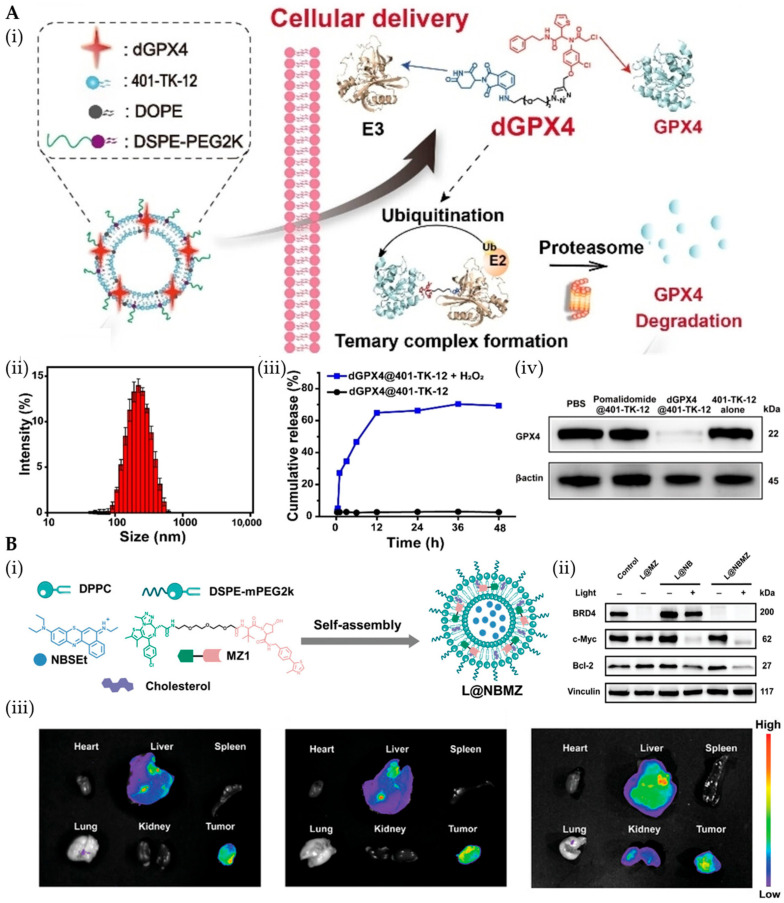
(**A**) (**i**) Schematic illustration of the intracellular delivery of dGPX4 and GPX4 degradation. (**ii**) Particle size distribution of dGPX4@401-TK-12. (**iii**) Drug release profile of dGPX4@401-TK-12. (**iv**) Western blotting detection of GPX4 levels in tumors after dGPX4@401-TK-12 nanoparticle treatment and other negative controls. Reprinted from Ref. [80], Copyright (2022), with permission from Wiley. (**B**) (**i**) Preparation scheme of L@NBMZ. (**ii**) Western blotting detection of BRD4, c-Myc, and Bcl-2 in MCF-7 cells. (**iii**) Fluorescence images of major organs and tumors at 10 h after intravenous injection of L@NR(Nile Red), L@NB, or L@NBMZ. Reprinted from Ref. [83], Copyright (2024), with permission from Wiley.

**Figure 4 pharmaceutics-17-01037-f004:**
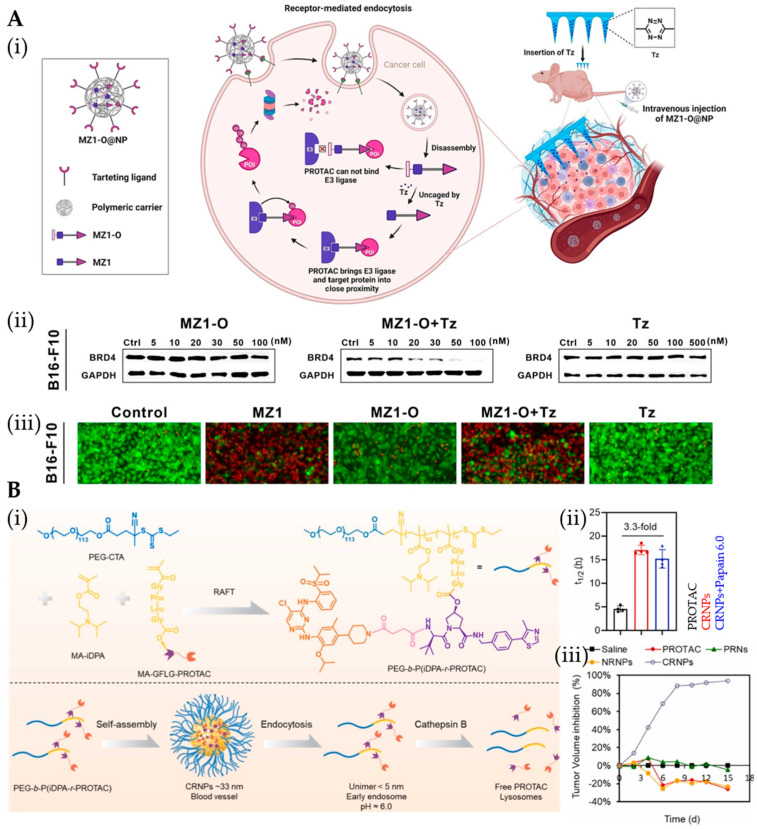
(**A**) (**i**) Illustration of the intracellular delivery of MZ1-O by a tumor-targeted polymeric carrier and the degradation of target proteins. (**ii**) Western blot analysis of BRD4 degradation upon the addition of various concentrations of MZ1-O or MZ1-O plus Tz (0.5 μM) in B16-F10 cells. (**iii**) LIVE/DEAD staining of B16-F10 cells treated with MZ1 (100 nM), MZ1-O (100 nM), MZ1-O (100 nM) plus Tz (0.5 μM), and Tz (0.5 μM). Reprinted from Ref. [120], Copyright (2023), with permission from Elsevier. (**B**) (**i**) Synthesis route of PEG-b-P(iDPA-r-PROTAC) polymer–drug conjugates. Schematic illustration of CRNPs self-assembly from polymer–drug conjugates and cascade responsiveness upon intravenous injection. (**ii**) Pharmacokinetic profiles (t_1/2_). (**iii**) Tumor volume inhibition rate. Reprinted from Ref. [44], Copyright (2023), with permission from Elsevier.

**Figure 5 pharmaceutics-17-01037-f005:**
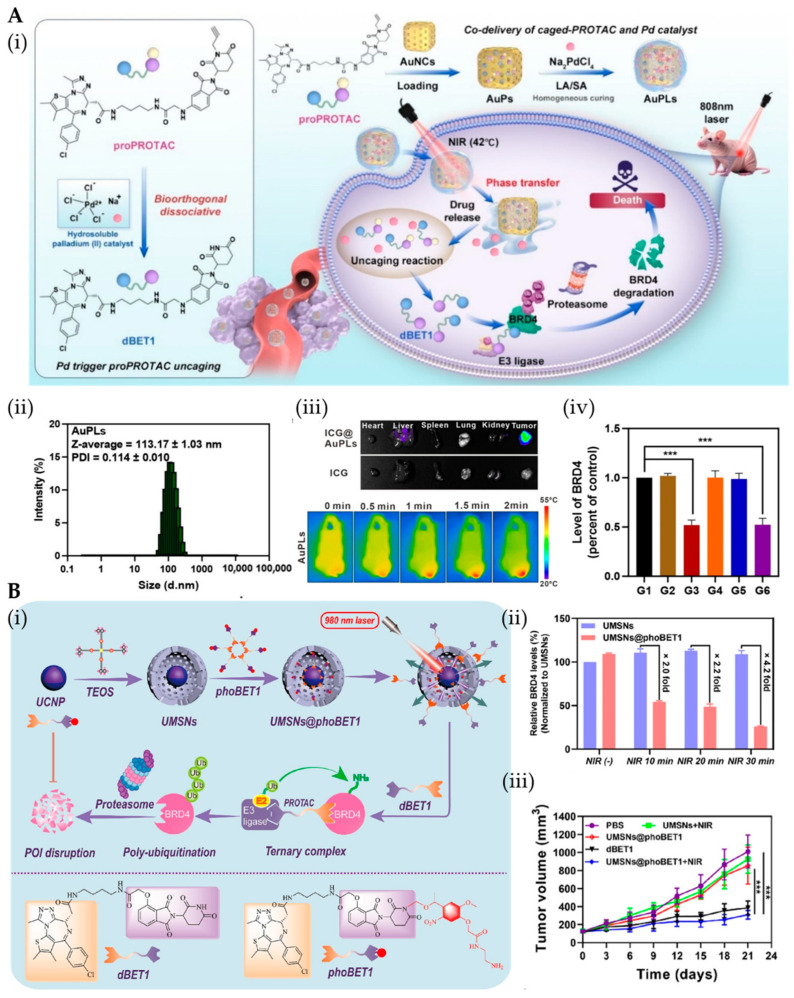
(**A**) (**i**) Photothermal-driven pro-prodrug platform of biorthogonal PROTACs uncaged through the Pd complex for co-delivery and bioorthogonal uncaging of the Pd (II)-catalyst for tumor-specific targeted treatment in breast cancer. (**ii**) The average size of the AuPLs is about 113.17 nm. (**iii**) Ex vivo fluorescence images of major organs and excised tumors and in vivo the photothermal effect of AuPLs accumulated within the tumor focal region. (**iv**) BRD4 protein expression levels were quantified by the software Image J. The data are presented as mean ± standard deviation. Statistical significance was determined using one- or two-way ANOVA analysis, with *** *p* < 0.001. Reprinted from Ref. [87], Copyright (2024), with permission from Elsevier. (**B**) (**i**) Schematic illustration for designing the NIR light-activatable PROTAC nanocage and its working mode for degrading the protein of interest (POI). (**ii**) Western blot analysis of BRD4 expression levels after different treatments. (**iii**) Curves of average tumor volume changes after the administration of different formulations during the 21 days treatment period (*n* = 5). Reprinted with permission from Ref. [88]; Copyright (2023) American Chemical Society.

**Figure 6 pharmaceutics-17-01037-f006:**
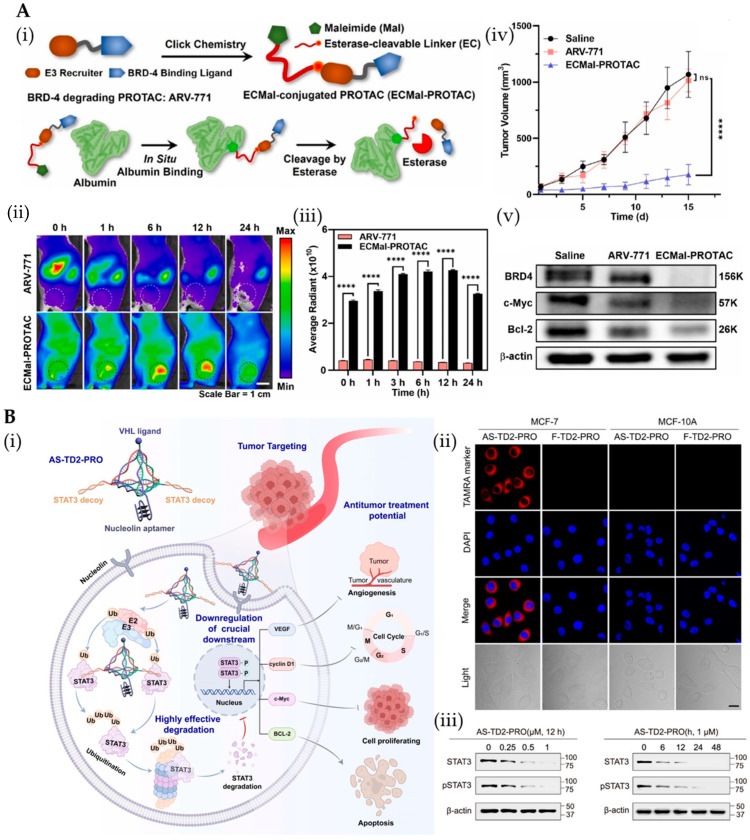
(**A**) (**i**) In situ albumin-binding esterase-cleavable BRD4-degrading PROTAC (ECMal-PROTAC) was developed by conjugating the ECMal linkers to ARV-771 molecules via two synthetic steps. (**ii**) The time-dependent biodistribution of Cy5.5-Alb-ECMal-PROTAC and Cy5.5-ARV-771. (**iii**) Their fluorescence intensities in tumor tissues were quantified (*n* = 3, **** *p* < 0.0001). (**iv**) The tumor growth profile according to the repeated treatment of saline, native ARV-771, or ECMal-PROTAC was obtained (*n* = 5, ns = not significant, **** *p* < 0.0001). (**v**) Western blot assay of excised tumor tissues with ECMal-PROTAC treatment showed reduced levels of BRD4, c-Myc, and Bcl-2. Reprinted from Ref. [89], Copyright (2023), with permission from Elsevier. (**B**) (**i**) Schematic diagram of the DNA tetrahedron-driven multivalent proteolysis-targeting chimera strategy. (**ii**) Confocal laser scanning micrographs of MCF-7 cells and MCF-10A cells after treatment with TAMRA-modified AS-TD2-PRO and F-TD2-PRO. (**iii**) WB analysis of the degradation of STAT3 and pSTAT3 in MCF-7 cells. Reprinted with permission from Ref. [161]; Copyright (2025) American Chemical Society.

**Figure 7 pharmaceutics-17-01037-f007:**
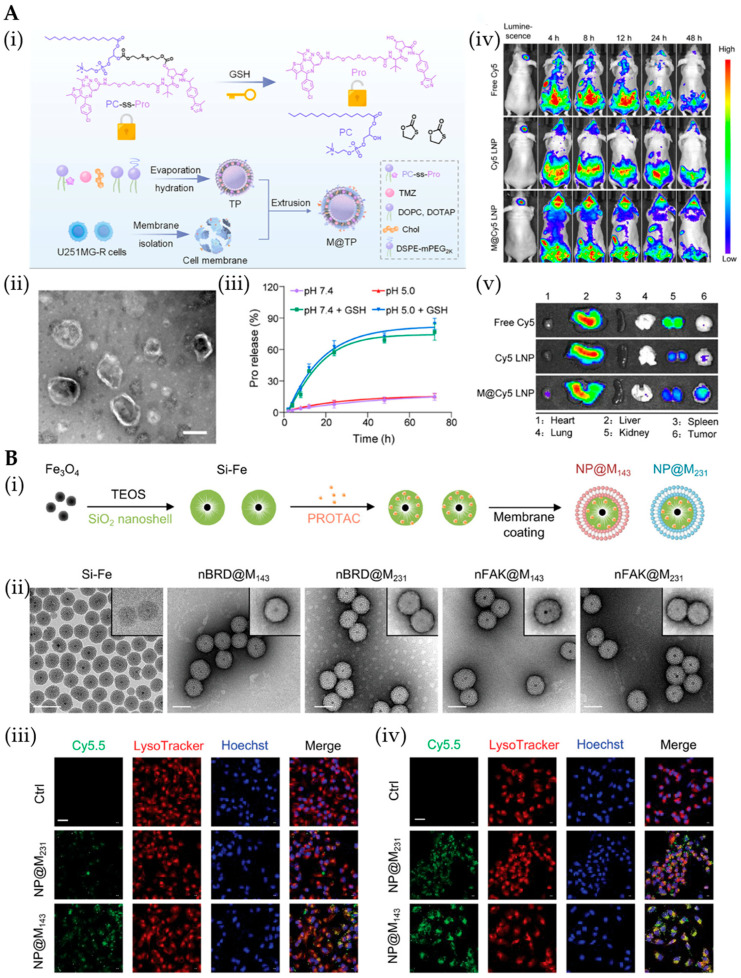
(**A**) (**i**) Schematic diagram of the fabrication of M@TP. (**ii**) TEM image of M@TP. (**iii**) Cumulative release of Pro from M@TP incubated with or without GSH (10 mm) at different pH values (7.4 or 5.0). (**iv**) In vivo distributions of free Cy5, Cy5 LNP, or M@Cy5 LNP in U251MG-R orthotopic GBM tumor-bearing mice at different time points post i.v. injection. (**v**) Ex vivo fluorescence images of major organs and glioma tumors from U251MG-R orthotopic GBM-bearing mice at 24 h after i.v. injection with free Cy5, Cy5 LNP, or M@Cy5 LNP. Reprinted from Ref. [93], Copyright (2025), with permission from Wiley. (**B**) (**i**) Schematic illustration of the preparation of NP@M143 or NP@M231. (**ii**) TEM images of Si-Fe, nBRD@M143, nBRD@M231, nFAK@M143, and nFAK@231. (**iii**) Confocal images of 143B tumor cells treated with Cy5.5-labeled NP@M143 or NP@M231 for 2 h. (**iv**) Confocal images of 143B tumor cells treated with Cy5.5-labeled NP@M143 or NP@M231 for 2 h in FMD medium. Reprinted from Ref. [163], Copyright (2025), with permission from Wiley.

**Table 2 pharmaceutics-17-01037-t002:** Application of nanotechnology for PROTACs.

Delivery System	PROTAC	POI	E3 Ligase	Loading Method	Release Behavior	Cancer/Cell Lines	Improvement	Ref.
Lipid NPs	dGPX4	GPX4	CRBN	Physical encapsulation	ROS responsive	HT-1080 cells	Improved tumor targeting and treatment safety	[80]
Galactosylated nanoliposomes	ARV-825	BRD4	CRBN	Physical encapsulation	Carrier dissociation	Hepatocellular carcinoma	Enhanced the uptake of tumor cells and the ability for active targeting	[81]
Folate-modified liposomes	DQ	NQO1	CRBN	Physical encapsulation	Carrier dissociation	A549 cells/lung cancer	Increased the solubility and the uptake of tumor cell	[43]
cRGD-modified cationic liposomes	inS3-TEG-VL	STAT3	VHL	Physical encapsulation (prodrug)	ROS responsive	Hepatocellular carcinoma	Enhanced the CSC-targeted delivery and lysosomal escape	[82]
Liposomes	MZ-1	BRD4	VHL	Physical encapsulation	Carrier dissociation	MCF-7/breast cancer	Improved the cell penetration and bioavailability of PROTAC	[83]
Polymeric micelle	ARV-825	BRD4	CRBN	Physical encapsulation	Carrier dissociation	Glioma	Enhanced the brain target ability	[48]
Polymeric NPs	ARV-771	BRD4	VHL	Physical encapsulation	GSH-responsive	Melanoma	Improved the pharmacokinetic profile of free PROTAC, facilitated accumulation in tumor cells	[84]
Polymeric NPs	ARV-771	BRD4	VHL	Self-assembly	Acid and light dual-activatable	Glioblastoma	Enhanced cellular uptake and accumulation at the tumor site	[49]
Nanomicelle	MZ-1	BRD4	VHL	Self-assembly	X-ray-responsive	Breast cancer	Accumulated at the tumor site owing to the EPR effect	[85]
Au nanoparticles	PROTAC peptide	GPX4	MDM2	Chemical conjugation	GSH responsive	Acute lymphoid leukemia	Facilitated intracellular action	[86]
Gold nanocages	dBET1	BRD4	CRBN	Physical encapsulation (prodrug)	Phase transfer	MCF-7/breast cancer	Precise control release in tumor tissue, leading to effective tumor accumulation	[87]
UCNP@mSiO_2_ NPs	dBET1	BRD4	CRBN	Physical encapsulation (prodrug)	NIR light-triggered	MV-4-11 cells	Spatiotemporal activation	[88]
Albumin-binding	ARV-771	BRD4	VHL	Chemical conjugation	Esterase-cleavable	4T1 cells/breast cancer	Improved pharmacokinetic properties and high accumulation in tumor tissue	[89]
Affibody-conjugate	MZ-1	BRD4	VHL	Self-assembly	GSH responsive	SKOV-3 cells/breast cancer	Accumulated in tumor site and then internalized by tumor cells	[50]
Camel milk-derived exosomes	ARV-825	BRD4	CRBN	Physical encapsulation	Carrier dissociation	Lung cancer cells	Conducive to stability and cellular uptake	[90]
PSI-coated gold-derived nanocluster	MP	MDMX	VHL	Chemical conjugation	pH/GSH responsive	Retinoblastoma/pancreatic cancer	Increased the blood-circulation time and tumor-specific accumulation	[91]
MOF-based nanoplatform	MZ-1	BRD4	VHL	Physical encapsulation	pH responsive	4T1 cells/breast cancer	Improved the intra-tumoral accumulation ability and prolonged their blood circulation time	[92]
Biomimetic hybrid nanovesicle	pro	BRD4	VHL	Physical encapsulation	pH/GSH responsive	U251MG-R/GBM	Homotypic tumor targeting capabilities and superior BBB penetration	[93]

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
