# Peer review of "Recent Advances in Nanomedicine: Cutting-Edge Research on Nano-PROTAC Delivery Systems for Cancer Therapy"

_pharmaceutics, 2025, doi:10.3390/pharmaceutics17081037_

Round 1

Reviewer 1 Report

Comments and Suggestions for Authors

The manuscript falls within the scope of the journal; however, it requires significant improvement based on the following comments:

Length: The manuscript is excessively long and should be reduced to approximately one-third of its current length to meet standard article expectations.

Content Originality: Much of the text is overly general and appears to repeat content from previous publications. A critical review is needed to remove redundant or repetitive sections and focus on novel insights.

Language and Terminology: There are several typographical errors and non-scientific or imprecise phrases that need revision. For example, the term "nano-delivery" should be replaced with more accurate terminology.

Concept Clarity (PROTAC): The concept of PROTAC is not clearly explained or sufficiently described, either in the main text or the abstract. It requires a clearer and more structured definition and discussion.

Reviewer 2 Report

Comments and Suggestions for Authors

Wu et al. have addressed an important and timely topic, presenting a comprehensive overview of recent advances in nano-PROTAC delivery systems aimed at improving pharmacokinetics, minimizing off-target effects, and enhancing therapeutic efficacy. The review is built upon a well-structured search strategy and successfully captures the major developments in the field, offering a valuable bird's-eye view for researchers and practitioners alike. The discussion is informative and accessible, catering to a broad scientific audience. However, several minor revisions could further enhance the quality and credibility of the manuscript. First, in many instances, the authors should include appropriate references to substantiate their claims and strengthen the academic rigor of the work. Just a few examples of many cases: 

  • Line 250-253: By incorporating hydrophobic or stimuli-responsive moieties, optimizing linker architectures, and leveraging coassembly for synergistic codelivery, researchers are progressively expanding the applicability and therapeutic potential of these systems.
  • Line 290-293: For example, studies have arranged multivalent E3 ligase ligands in polymeric arrays, forming multisite recruitment scaffolds on nanoplatforms via self-assembly to increase the binding stability and degradation efficiency of the ternary complex.
  • Table one lacks references.
  • Line 313-318: NDDSs have been extensively utilized for the delivery of diverse antitumor agents because of their favorable biocompatibility, structural tunability, high drug loading efficiency, and capacity for controlled release within complex physiological environments. With the successful application of nanodelivery platforms for small-molecule drugs, nucleic acid therapeutics, and protein-based pharmaceuticals, researchers have progressively integrated nanocarrier technology into PROTACs delivery.

Second, a comparative analysis of the various delivery strategies in the conclusion section, particularly with respect to their therapeutic outcomes, would add significant value. Such an analysis could help find out the most promising approaches for future research.

Reviewer 3 Report

Comments and Suggestions for Authors

In this manuscript titled “Recent advances in nanomedicine: cutting-edge research on nano-PROTAC delivery systems for cancer therapy”, Wu et al. discuss how integration of PROTACs into nanocarriers (nano-PROTACs) can help in overcoming various issues that currently hinder the widespread clinical translation of PROTACs. The topic is a rapidly emerging field of great interest. As such, the review is timely and informative. However, I feel the review can be improved, as I detail below:

1, Overall, there are numerous places in the manuscript that go without references. As references not only support the claims made but also is an educational resource for introductory readers, I believe that making it easy to identify which claim is supported by which reference is very important.

2, lines 48-51: The two claims made here, PROTACs 1) expanding the druggable space and also 2) achieving degradation at substoichiometric doses should be supported by appropriate references.

3, lines 51-53: Mutations in the E3 ligases or the binding site of the POI can cause resistance to PROTACs, so I am not really sure about this claim. Perhaps the authors could provide a reference or consider rephrasing?

4, line 83: I think “nanoDDS” is a more common abbreviation than “NDDS”.

5, Figure 2: The figure is informative, but I think the design could be improved. The relatively large amount of text within the figure somewhat obscures the graphical component. Since the accompanying Table 1 discusses most of the textual info here, perhaps the authors can consider reducing/omitting the explanation by text here? Also, given the detailed discussion in 2.1 onwards, perhaps each strategy of nanoPROTAC design can be elaborated further in the figures?

6, Line 289: Some readers may not necessarily know the “hook effect”. Perhaps a short explanation may be helpful.

7, Section 2: For each of the nanoPROTAC design strategies, the authors briefly mention some future avenues of research. Not really a strong request, but if I feel that if the authors to elaborate a little but more this would strengthen their review (for example at the end of section 2.4 the authors mention “advancements in molecular self-assembly technologies, ..., etc.” If the authors can mention some specifics regarding what kind of advancements are necessary to overcome current limitations, this would significantly enhance the review).

8, Lines 313-336: The text here is somewhat repetitive of former sections and could probably be reduced.

9, Lines 337-339: “Upward trajectory”—please detail. Do the authors mean the number of publications? The quality of the nanoPROTACs? Etc.

10, Section 3: I sometimes felt some of the studies were explained in too much detail. Please check whether the details provided are necessary for the purposes of this review, and shorten if possible.

11, Section 3.4: I am not sure what the authors mean by “biotechnology carriers”. Considering the examples here are albumin, antibodies, etc., I imagine “biological” or “bio-inspired” might be more suitable.

12, Sections 2/3: Section 2 explains nanoPROTACs by construction, Section 3 by material/platform, as illustrated in Figure 1. However, the sections as is are relatively independent of each other, without much integration. Would it be possible for the authors to discuss the relationship between, and integrate, sections 2 and 3?

13, Table 2: The table is truncated at line 932.

14, Section 4: The authors might want to discuss the fact nanoPROTAC strategies lead to a increase in size, and that certain tumors require stringent control of size to be able to permeate in the first place.

Round 2

Reviewer 1 Report

Comments and Suggestions for Authors

all my comments were addressed

Author Response

Thank you very much for your suggestions on the article.Taking into account the opinions of the other reviewers, we have revised the manuscript again for your review.

Reviewer 3 Report

Comments and Suggestions for Authors

Thank you for the revisions. I felt the authors have sufficiently addressed most of the points I raised in my previous review. One thing that I still feel that could be improved, is that while references have been added, I personally would prefer placing references more frequently (not necessarily more papers/references, just clarifying which claim/sentence is supported by which paper/reference).

Author Response

We would like to thank Reviewer 3 for these constructive comments which are of great help to enhance the quality of our manuscript. We have carefully revised the manuscript according to your comments. Below are our point-by-point responses to your comments and the corresponding revisions in track changes in the re-submitted files.

Comments 1: Thank you for the revisions. I felt the authors have sufficiently addressed most of the points I raised in my previous review. One thing that I still feel that could be improved, is that while references have been added, I personally would prefer placing references more frequently (not necessarily more papers/references, just clarifying which claim/sentence is supported by which paper/reference).

Response 1: Thank you for pointing this out. We agree to include appropriate references as suggested. Therefore, we have incorporated additional references to substantiate key points, and expanded the literature coverage for greater rigor. In this revised manuscript, I have added appropriate references to support my arguments. You can check the changes made by me by tracking.
